# Evaluating Methane Adsorption Characteristics of Coal-Like Materials

**DOI:** 10.3390/ma13030751

**Published:** 2020-02-06

**Authors:** Pengxiang Zhao, Hui Liu, Chun-Hsing Ho, Shugang Li, Yanqun Liu, Haifei Lin, Min Yan

**Affiliations:** 1College of Safety Science and Engineering, Xi’an University of Science & Technology, Xi’an 710054, China; zhpxhs@sina.com (P.Z.); lhui0606@163.com (H.L.); lisg@xust.edu.cn (S.L.); selina81256556@126.com (Y.L.); lhaifei@163.com (H.L.); minyan1230@xust.edu.cn (M.Y.); 2Key Laboratory of Western Mine Exploitation and Hazard Prevention Ministry of Education, Xi’an University of Science and Technology, Xi’an 710054, China; 3Department of Civil Engineering, Construction Management and Environmental Engineering, Northern Arizona University, Flagstaff, AZ 86011, USA

**Keywords:** adsorption materials, multiple factors, the ratio of methane adsorption, adsorption principle

## Abstract

In order to investigate the methane adsorption characteristics of coal seam materials in a “solid–gas” coupling physical simulation experiment, activated alumina, silica gel, the 3Å molecular sieve, 4Å molecular sieve and 5Å molecular sieve were selected as adsorption materials. According to the pore structure and adsorption characteristics, coal samples at the Aiweiergou #1890 working face were prepared as compared materials. The WY-98A methane adsorption coefficient measuring instrument was used to carry out this adsorption experiment under different temperatures, particle sizes and moisture contents. The results suggested that the adsorption principles of three kinds of molecular sieves under multiple factors do not fully fit a Langmuir adsorption model, and cannot be used as adsorption materials. The changing trend of the adsorption increment of activated alumina and silica gel are similar to that of coal samples, so they can be used as a coal-like materials. The methane adsorption coefficient *a* value changing trends of activated alumina and silica gel appear to be the same as the Aiweiergou #1890 coal samples, but the results from silica gel are closer to that of coal samples. Thus, silica gel is preferred as the adsorption material. The result provides an experimental basis for the selection of methane-adsorbing materials and carrying out “solid–gas” coupling physical simulation experiments in a physically similar testing model.

## 1. Introduction

With an increasing mining depth of coal extraction, there have been growing concerns in coal and methane outbursts and mining safety in general. To study the “solid–gas” coupling phenomenon using a physical-similar model is the most effective laboratory method to simulate gas drainage at mining sites. Selecting the material with the same adsorption characteristics as the original coal and having appropriate methane adsorption properties are the key to carry out the test. In recent years, many scholars have carried out the principles and mechanical analysis of the methane adsorption of raw coal from various angles of the seam. For example, a group of scholars [1,2,3,4] studied the methane adsorption principle of coal under high pressure. Their research works indicated that as the pressure increases, the amount of adsorption increases as well, although the rate of increase becomes slowed down. In addition, the amount of methane adsorption decreases slightly with the increase of pressure after 8 MPa.

Many researchers [5,6,7] and other experiments through the methane adsorption of coal containing several different particle sizes concluded that the methane adsorption amount increases as the coal particle size decreases within a certain range. Some researchers [8,9,10,11,12,13] used the artificial humidification method to study the methane adsorption principle of coal samples under different moisture contents. Their findings showed that the rate of methane adsorption decreased with the increase of moisture content based on the microscopic mechanism. The effect of the temperature on the methane adsorption was evaluated by some scholars [14,15,16,17], and their studies suggest that coal adsorption is an exothermic process, so that as the temperature increases, the amount of methane adsorption decreases. Research works [18,19] showed that the adsorption characteristics of coal-like material vary, depending on several factors including its temperatures, particle sizes and moisture contents. These factors are found to be in compliance with the Langmuir monolayer adsorption model.

At present, in order to investigate the coupling principle of mining cracks evolution and methane adsorption desorption, many scholars [20,21] have studied the materials of a “solid–gas” coupling physical simulation experiment. Given that the raw coal is not very convenient to obtain from coal mines, it is necessary to maintain a similarity of the methane adsorption property between the coal-like materials and the raw coal, while achieving the similarity requirement of the compressive strength under the small molding pressure. Therefore, this paper presents a series of “solid–gas” coupling physical simulation experiments (including methane adsorption experiments) using coal-like materials to evaluate the properties of coal-like materials, including silica, activated alumina, the 3Å molecular sieve, 4Å molecular sieve, 5Å molecular sieve, and other materials. In addition, three factors (the particle size of the coal sample, mining temperature and the moisture of coal materials) were taken into the account for the analysis. The results of experiments provide a better understanding for the application of coal-like materials in the “solid–gas” coupled physical simulation and analysis.

## 2. Device and Experimental Design

### 2.1. Experimental Materials

The “solid–gas” coupling physical simulation experiment not only requires that the methane adsorption characteristics of coal-like materials meet the similarity criteria, but that they also meet the general physical and mechanical properties. In this regard, the coal-like material components are selected in a way that the properties of specimens produced in the laboratory would be similar to the coals at the mining site. Therefore, a large amount of research and literature [22,23,24,25] was carried out to better understand the performance of coal-like materials. According to the appearance and pore characteristics of solid particles, and the adsorption of other gases, the following five components including activated alumina (α-Al_2_O_3_), silica gel (mSiO_2_·nH_2_O) and three kinds of molecular sieve (the general chemical composition is (M)_2/n_O Al_2_O_3_·xSiO_2_·pH_2_O): the 3Å molecular sieve, the 4Å molecular sieve and the 5Å molecular sieve, were selected (as shown in Figure 1) in the experiment.

### 2.2. Experimental Design

#### 2.2.1. Different Influencing Factors

(1) Particle sizes

According to the requirements of the instrument, and suggestions of previous research work [26], five sets of sieves (#20, #40, #60, #80 and #100) were used to prepare aggregates for the methane isothermal adsorption test.

(2) Temperatures

According to the relevant research on the influence of temperature on coal adsorption characteristics [27], four different temperatures of 20, 30, 40 and 50 °C (i.e., the temperature gradient of 10 °C) were selected for the methane isotherm adsorption test.

(3) Moisture contents

Four moisture contents (by weight, 0%, 2%, 5% and 10%) were used to prepare the mixtures using the self-developed coal sample humidifier for the isothermal adsorption test [28].

#### 2.2.2. Experimental Design and Steps

(1) Pore structure test

The pore distribution of various test materials was measured by liquid nitrogen adsorption method using ASAP2020 surface area and porosity analyzer. The adsorption and desorption curves were obtained under different pressures. The specific surface areas of specimens were calculated according to the Brunauer, Emmett and Teller (BET) [29] multilayer adsorption theoretical model, and the pore volumes of specimens were calculated using the Barrett, Joyner and Halenda (BJH) [30] aperture distribution theoretical model.

(2) Isothermal adsorption test

The true density and apparent density of coal samples were measured according to the GB/T 217-2008 standard (measurement method of the true density of coal) [31]. The standard specifies the standard determination method of true and apparent density. Sodium dodecyl sulfate solution is used as the infiltrating agent to make the sample infiltrate and settle in the density bottle. Exclude the adsorbed methane, and calculate the relative density of the sample based on the mass of the same volume of water excluded by the sample. After completion of sieve analysis, the aggregate retained on each sieve was placed in a vacuum oven at 60 °C for 6 hours. After being dried, the aggregates were quickly weighed and labeled as m_1_, and then the varying amounts of water were added to the aggregates based on the designed moisture content (0%, 2%, 5% and 10% by weight) using the coal sample humidifier. Subsequently, all materials are quickly transferred to the sample tanks of WY-98A to perform the isothermal adsorption test determine. The adsorption pressure, degassing time, adsorption time and temperature were recorded prior to the experiment.

## 3. Results and Analysis

### 3.1. Pore Structure Results

The specific surface areas and pore volume distributions of the test materials (activated alumina, silica gel, 3Å molecular sieve, 4Å molecular sieve, 5Å molecular sieve and the Aiweiergou #1890 working face coal sample) were obtained according to the nitrogen adsorption test. The test results are shown in Figure 2, where the Aiweiergou #1890 working face coal sample, activated alumina, silica gel, 3Å molecular sieve, 4Å molecular sieve and 5A molecular sieve are labeled as coal, AA, SG, 3Å, 4Å and 5Å.

B.B. Hoodot [32] divided the pores in coal into micropores (<10 nm), transition pores (10–100 nm), mesopores (100–1000 nm) and macropores (>1 μm). The pore volume of the nanoscale pores (0–100 nm) determines the adsorption rate of coal, and the specific surface area of the nanoscale pores determines the ultimate adsorption of coal [33]. Moreover, the macroporous adsorption effect of diameter greater than 1 μm is not obvious; therefore, only micropores, transition pores and mesopores are considered in this test.

### 3.2. The Test Results of Adsorption Coefficients

#### 3.2.1. The Test Result of Adsorption Coefficient

The adsorption coefficient *a* of each material prepared at different temperatures (20, 30, 40 and 50 °C) and four moisture contents (0%, 2%, 5% and 10%) were obtained according to the isothermal adsorption test. The test results are shown in Figure 3.

The adsorption coefficient *a* has different fluctuations under different factors. The controlling variable method was adopted in the experiment. In Figure 3a, the designed particle size was 60–80 mesh, the moisture content was 0%. Compared with three molecular sieves, with the increase of temperature, the adsorption coefficient *a* values of the #1890 working face coal sample, activated alumina and silica gel, vary slightly. In addition, the adsorption coefficient *a* value of 4Å molecular sieve is negative. As shown in Figure 3b, the designed temperature was 30 °C and the moisture content was 0%. The adsorption coefficient *a* values of the #1890 working face coal sample, activated alumina and silica gel increase with the increase of particle size, and the adsorption coefficient *a* values of the three molecular sieves fluctuate considerably. When it comes to moisture content, as shown in Figure 3c, the designed temperature was 30 °C and the particle size was 60–80 mesh. The adsorption coefficient *a* values of the #1890 working face coal sample, activated alumina and silica gel showed no obvious downward trend. However, the adsorption coefficient *a* values of the molecular sieves fluctuated greatly, and the adsorption coefficient *a* value of the 3Å molecular sieve showed negative values twice. Since the value of the adsorption constant *a* is the result of Langmuir model fitting according to the equilibrium methane adsorption capacity at different pressure points, it can be seen from the adsorption curve that the adsorption curve of the molecular sieve under different factors is different from that of the Langmuir adsorption model, so the adsorption result is negative.

#### 3.2.2. The Test Result of Adsorption Coefficient B Values

For the above five materials (AA, SG, 3Å, 4Å and 5Å) and the Aiweiergou #1890 working face coal sample, according to the isothermal adsorption test, the adsorption coefficient *b* values of each material under different influencing factors were obtained. The test results are shown in Figure 4.

The adsorption coefficient *b* value varied based on temperatures, particle sizes and moisture contents. Under the effect of the three factors, it is found that the adsorption coefficient *b* values of the #1890 working face coal sample in Aiweiergou are between 0.3 MPa^−1^ and 0.61 MPa^−1^, the adsorption coefficient *b* values of activated alumina are between 0.057 MPa^−1^ and 0.066 MPa^−1^, and the adsorption coefficient *b* values of silica gel are between 0.114 MPa^−1^ and 0.194 MPa^−1^. The coefficient *b* values of the coal sample are large as shown in Figure 4a–c. In Figure 4a, the designed particle size was 60–80 mesh, the moisture content was 0%. The coefficient *b* values of activated alumina and silica gel consistent with the change law of coal sample. The coefficient *b* values of the coal sample, activated alumina and silica gel have no significant change in Figure 4b, the designed temperature was 30 °C and the moisture content was 0%. In Figure 4c, the designed temperature was 30 °C and the particle size was 60–80 mesh. In addition, the coefficient *b* values of the 3Å molecular sieve showed negative values twice. Obviously, the adsorption coefficient *b* value is highly dependent on the three factors.

### 3.3. Adsorption Isotherm Test Results

#### 3.3.1. Test Results of Adsorption Isotherms of Test Materials at Different Temperatures

The WY-98A was used to measure the methane adsorption capacity of coal-like materials, such as silica gel, activated alumina, the 3Å molecular sieve, 4Å molecular sieve and 5Å molecular sieve at the four designed test temperatures, at the moisture content of 0% and under different design pressures. The methane adsorption isotherms of various materials are shown in Figure 5.

It can be seen from Figure 5 that the methane adsorption process of the Aiweiergou #1890 working face coal sample at different temperatures exhibits a “steep-slow”-shaped adsorption isotherm. After a sharp rise, and the amount of adsorption decreases with the increasing temperature. The methane adsorption isotherms of activated alumina at different temperatures show a linear trend, and the adsorption amount decreases with this increasing temperature. The methane adsorption isotherms of silica gel at different temperatures show a gentle trend after a sharp rise, but the adsorption decreases with the increase of temperature. The adsorption isotherms of 3Å molecular sieves, 4Å molecular sieves and 5Å molecular sieves at different temperatures showed a steep-slow-steep rise trend. As the temperature reached to 30 °C, the amount of adsorption of the three molecular sieves was small after 4 MPa, indicating that the entire adsorption process is also in line with the “steep-slow” upward trend. The methane adsorption capacity value of each molecular sieve at different temperatures is different, and has no obvious pattern.

#### 3.3.2. Test Results of Adsorption Isotherms of Test Materials at Different Particle Sizes

The determinations of the methane adsorption capacity of the Aiweiergou #1890 working face coal sample, activated alumina, silica gel, 3Å molecular sieve, 4Å molecular sieve and 5Å molecular sieve, are performed by WY-98A. The moisture content of 0% and the constant temperature of 30 °C were used in the test. The methane adsorption capacity of different particle sizes under different design pressure conditions is obtained. The methane adsorption capacities of various materials are shown in Figure 6.

It can be seen from Figure 6 that the methane adsorption capacities of the five testing materials are similar with the ones shown in Figure 5. The methane adsorption isotherms of the coal samples collected from the #1890 working face in Aiweiergou, and the methane adsorption capacity of the activated alumina and silica gels, increased as the pressure increases. However, the adsorption isotherms of coal-like materials sampled from 3Å molecular sieves, 4Å molecular sieves and 5Å molecular sieves showed different trends as compared to the previous three material samples. The findings provide the fact that the characteristics of the coal-like materials collected from 3Å molecular sieves, 4Å molecular sieves and 5Å molecular sieves exhibit different mechanical behavior. Specifically, when the size of the 3Å molecular sieve was retained between #20–#40 sieves, #40–#60 sieves and #60–#80 sieves, the 4Å molecular sieve at #60–#80 sieves, and the 5Å molecular sieve at #40–#60 sieves and #60–#80 sieves, the methane adsorption capacity of the three kinds of molecular sieve after 4 MPa changes slightly, considering that the whole adsorption process is in line with the rising trend of "steep-slow". The methane adsorption capacity value of the three kinds of molecular sieve was different under different particle sizes, and has no important pattern. 

#### 3.3.3. Test Results of Adsorption Isotherms of Test Materials with Different Moisture Content

The methane adsorption capacities of the #1890 coal sample, activated alumina, silica gel, 3Å molecular sieve, 4Å molecular sieve and 5Å molecular sieve in the #60–#80 sieves particle size were obtained using the experimental design moisture contents, constant temperature 30 °C, and different design pressure conditions. The measurement was carried out by the WY-98A. The methane adsorption capacity for each material are shown in Figure 7.

It can be seen from Figure 7 that the effect of the moisture content of coal materials in the adsorption isotherm of the methane adsorption is significant for the samples collected from the Aiweiergou #1890 working activated alumina and silica gel. The adsorption isotherms of 3Å molecular sieves, 4Å molecular sieves and 5Å molecular sieves at different moisture contents fluctuated differently among these three materials. It can be seen that when the moisture content of the 3Å molecular sieve is at 0%, the moisture contents of 4Å molecular sieve are at 0%, 5% and 10%, and the moisture contents of the 5Å molecular sieve are at 0%, 2% and 5%, the amount of adsorption of the three molecular sieves tend to be is less after 4 MPa. In addition, there was no significant change in the adsorption capacity of each molecular sieve under different pressures.

## 4. Discussion

### 4.1. Analysis of Pore Structure Results and Langmuir Model Fit

#### 4.1.1. Analysis of Pore Structure Results

It is known from Figure 2a that the pore volume of silica gel and alumina in micropores is much larger than that of other test materials, and the pore size distribution of silica gel and alumina is concentrated in the nanometer pores. The pore volume of alumina in micropores is 2788 times and 11 times higher than that of the coal and molecular sieve, respectively. The pore volume of silica gel in micropores is 3007 times and 12 times higher than that of the coal and molecular sieve, respectively. The materials of the three molecular sieves and the pore volume of the transition pore are slightly higher than that of the micropores and mesopores.

Figure 2b shows that the Specific Surface Area (SSA) of the test materials with different pore sizes also showed different trends. The larger the pore size, the smaller the SSA of the test material. At the same pore size, the specific surface area of silica gel and alumina is greater than the specific surface area of three molecular sieves of 3Å, 4Å and 5Å.

#### 4.1.2. Langmuir Model Fitting Analysis

The Langmuir adsorption model is a monolayer adsorption model. Since the isotherm of coal adsorption methane conforms to the first type of isotherm classified by the International Union of Pure and Applied Chemistry (IUPAC) adsorption isotherm, researchers use the Langmuir adsorption model to calculate the amount of coal methane adsorption [34,35,36]. The expression is:(1)$$Q=\frac{abp}{1+bp}$$
where: Q is the methane adsorption amount, in mL/g; p is the gas pressure, in MPa; a is the adsorption coefficient, indicating the limit adsorption amount during the methane adsorption process, given in m^3^/t; b is the adsorption coefficient, which indicates the methane pressure at half the limit adsorption amount, and it is in MPa^−1^.

The Langmuir model was fitted by the adsorption isotherms of activated alumina, silica gel, the 3Å molecular sieve, 4Å molecular sieve and 5Å molecular sieve; the fitting results are shown in Table 1.

It can be seen from Table 1 that the adsorption isotherm of the #1890 coal material, activated alumina and silica gel under the influence of the three factors show well agreement with the Langmuir model, provided the fitting results are greater than more than 0.95. The findings confirm the roles played by the three factors (temperature, particle size and moisture) have a significant influence on the adsorption properties of coal-like materials. However, the fitting results of the three molecular sieves are not well explained with the Langmuir model, indicating the fact that the three factors (temperature, particle size and moisture) might not have a significant impact on the properties of the adsorption of coal-like materials.

### 4.2. Analysis of Ultimate Adsorption Capacity and Adsorption Properties of Materials

Coal-like materials are considered as a part of adsorbent materials due to the basic characteristics that the coal-like materials have a large as adsorption capacity similar to the coal. From the results of adsorption isotherm tests under the three factors, it can be seen that the methane adsorption capacity of coal-like materials perform well at the conditions of a fixed temperature of 30 °C, a moisture content of 0%, and a particle size of the #60–#80 sieve. According to the fitting results of the Langmuir model, the methane adsorption properties of the three molecular sieves under different influencing factors is quite different from that of the original coal. Therefore, the multi-factor analysis of the methane adsorption capacity of the material is only for silica gel and activated alumina.

#### 4.2.1. Comparative Analysis of Material and Ultimate Adsorption Capacity of Coal Methane

In the solid–gas coupling physically similar simulation experiment, the methane adsorption characteristics of similar materials in the simulated coal seam are mainly based on the methane adsorption amount, and the *a* value is the ultimate adsorption amount of the material; therefore, the adsorption coefficient *a* value is selected for comparison purposes.

It can be seen from Figure 8 that the adsorption coefficient *a* values of the activated alumina, the silica gel and the Aiweiergou #1890 working face coal sample, all vary. The adsorption coefficient *a* of the #1890 working face coal sample in Aiweiergou is relatively small, and its value is 41.194 m^3^/t. The methane adsorption coefficient *a* values of activated alumina and silica gel are relatively close, and their values are 70.721 and 77.699 m^3^/t, respectively, which is more than 1.5 times greater than that of Aiweiergou #1890 working face coal sample. It can be seen from Figure 2 that since the specific surface area of the micropores of the silica gel and the activated alumina is large, more adsorption sites can be provided for the methane molecules in the methane adsorption process. Additionally, the van der Waals force exists in the surface of the methane and the pore surface of the material, and the larger specific surface area can increase the contact between the methane molecule and the pore surface of the material. Therefore, a large force is generated, so the adsorption constant *a* is increased.

Because the coal-like materials are developed by mixing the existing cement with the adsorbing material, the addition of materials such as cementing agent will affect the adsorption of the whole material, so the material with larger adsorption capacity than the raw coal has a large adjustment range. Therefore, the coal-like materials meet the requirements of the adsorbent material in terms of the amount of adsorption. 

#### 4.2.2. Analysis of Material and Coal Methane Adsorption Change Properties

In order to more explicitly express the trend of the methane adsorption properties of the testing materials, each pressure point is divided based on the methane adsorption amount of the coal sample collected at the #1890 working face of the Aiweiergou. After this, the ratios of methane adsorption capacity of the #1890 coal sample versus to the activated alumina and silica gel were calculated, and the results are shown in Figure 9.
(2)$$y=3.306x^{-0.304}$$
(3)$$y=1.247x^{-0.191}$$

The ratio curve of methane adsorption capacity of between the silica gel/activated alumina and the Aiweiergou #1890 working face coal sample indicate a good relationship between the absorption capacity and the pressure are shown in Figure 9. The fitting formulas are shown in Equations (2) and (3). It can be seen from the formula, that as the pressure increases, the ratio between the #1890 coal sample and activated alumina and the ratio of between the #1890 material and silica gel decreases. It is indicated that with the increase of pressure, the methane adsorption capacity of the material is greater than the methane adsorption capacity of the coal sample.

### 4.3. Analysis of Factors Affecting the Ultimate Adsorption of Materials

#### 4.3.1. Analysis of the Influence Principles of Material Ultimate Methane Adsorption 

The adsorption coefficient *a* value characterizes the ultimate adsorption capacity of the material. The variation of the adsorption coefficient *a* of various materials (#1890 coal sample, silica gel and activated alumina) with various factors (temperature, particle size and water content) is shown in the Figure 10.

It can be seen from Figure 10a that hydrogen bonds can be formed with water molecules due to the existence of many polar dangling bonds on the coal surface. Then, because of the intermolecular force, the coal sample has strong water absorption, and the adsorbed water will occupy the pores of some coal. Therefore, with the increase of moisture content, the methane adsorption capacity of the #1890 working face coal sample in Aiweiergou gradually decreases. Since the decrease of the particle size increases the specific surface area of the coal [37], the increased specific surface area provides more adsorption sites for methane adsorption, so the methane adsorption amount increases as the particle size decreases. Since coal adsorption methane is an exothermic process [38,39], the increase in temperature inhibits the adsorption of methane by coal, so the amount of adsorption decreases. The value of the adsorption coefficient *a* decreases parabolically with the change of moisture content and particle size, and decreases linearly with the increase of temperature.

It can be seen from Figure 10b that as the moisture content increases, the methane adsorption capacity of the silica gel decreases as a power function. The fitting formula is shown in Table 2. This is because the functional group (hydroxyl, a structural polar group very close to water) on the surface of the silica gel molecule has strong hydrophilicity that leads to strong water absorption. The adsorption coefficient *a* decreases as the particle size and temperature increase in the parabolic and linear forms, meaning that the exothermic process of silica methane adsorption caused the decrease of the particle size, as well as the increase in the specific surface area.

It can be seen from Figure 10c that as the moisture content, temperature and particle size increase, the adsorption coefficient *a* of the activated alumina decreases. The change principle is shown in Table 2.

This variation is similar to that of the #1890 coal sample in Aiweiergou. However, from the analysis of the adsorption coefficient *a* value of materials and coal samples from the three factors, that is, moisture, particle size and temperature, the adsorption coefficient *a* value of silica gel has a higher similarity under the three factors than that of activated alumina.

#### 4.3.2. Sensitivity Analysis of Methane Adsorption Ultimate of The Coal-Like Material

The range analysis method can explicitly see the fluctuation of different levels of indicators under the influence of each factor, which uses the maximum value of each influencing factor to subtract the minimum value. According to the size of fluctuation range, it can be clearly seen which factor has a greater influence on the index, that is, it is more sensitive to a factor. Since the adsorption coefficient *a* is the ultimate adsorption capacity of the material, therefore, the value of a is chosen to measure the multi-factor sensitivity of the methane adsorption ultimate of materials. (1, 2, 3 and 4 in the table indicate different factors from low to high, respectively).

The methane difference adsorption a value of the #1890 working face coal sample in Aiweiergou is analyzed by the range of the three factors. The analysis results are shown in Table 3. It can be seen from the results of the range analysis that the temperature has the greatest influence on the adsorption coefficient *a* of the #1890 working face coal sample in Aiweiergou, followed by the particle size, and the smaller is the water. The sensitivity of the three factors to the influence of the adsorption coefficient *a* value of the #1890 working face coal sample in Aiweiergou is: temperature > particle size > moisture content.

The methane adsorption *a* value of the activated alumina at different condition of the three factors was analyzed by the range analysis method. The analysis results are shown in Table 4. It can be seen from the results of the range analysis that the particle size has the greatest influence on the adsorption coefficient *a* value of the activated alumina, followed by the temperature. The sensitivity of the three factors to the influence of the adsorption coefficient *a* value of activated alumina is: particle size > temperature > moisture content.

The range analysis of the adsorption coefficient *a* of silica gel at different levels of various factors was carried out, and the results are shown in Table 5. It can be seen from the results of the range analysis that the particle size has the greatest influence on the adsorption *a* value of silica gel, followed by the moisture content; the sensitivity of the three factors to the adsorption of activated alumina is: particle size > moisture content > temperature.

The sensitivity of each factor to the properties of raw coal methane adsorption is: temperature > moisture content > particle size. The range analysis of activated alumina and silica gel under the three factors shows that moisture content, particle size and temperature have some influence on the methane adsorption capacity of the activated alumina and silica gel.

## 5. Conclusions

Through the tests and analyses on coal-like specimens, the paper has the following conclusions:
(1)The adsorption process of silica gel and activated alumina under the influence of particle size, temperature and moisture have good agreement with the Langmuir adsorption model; it is consistent with the methane adsorption properties of raw coal under the three different factors. The adsorption processes of the three molecular sieves under the influence of particle size, temperature and moisture are quite different, and seemed not to possess a good fit with the Langmuir adsorption model; they cannot be used as adsorption materials.(2)The specific pore surfaces of the micropores and transition pores of silica gel and activated alumina are large, so the adsorption coefficient *a* has a large value. As the pressure increases, the methane adsorption ratio gradually decreases and then it tends to be stabilized. The adsorption amount and the change of the adsorption capacity of different pressures are in line with the methane adsorption characteristics of coal-like materials simulating raw coal.(3)Temperature, particle size and moisture content have a significant influence on the methane adsorption of the #1890 working face coal sample in Aiweiergou, the silica gel and activated alumina. However, the sensitivity of the materials to different factors is different. By analyzing the change trend of the adsorption amount of the #1890 working face coal sample in Aiweiergou, silica gel and activated alumina, the change trend equation is obtained. 

The variation trends of the three test materials under different factors are the same, but the variation of the methane adsorption capacity of silica gel under different factors is similar to that of the #1890 working face coal sample in Aiweiergou. Based on the sensitivity analysis, this silica gel is more suitable as adsorption material than the other two materials.

## Figures and Tables

**Figure 1 materials-13-00751-f001:**
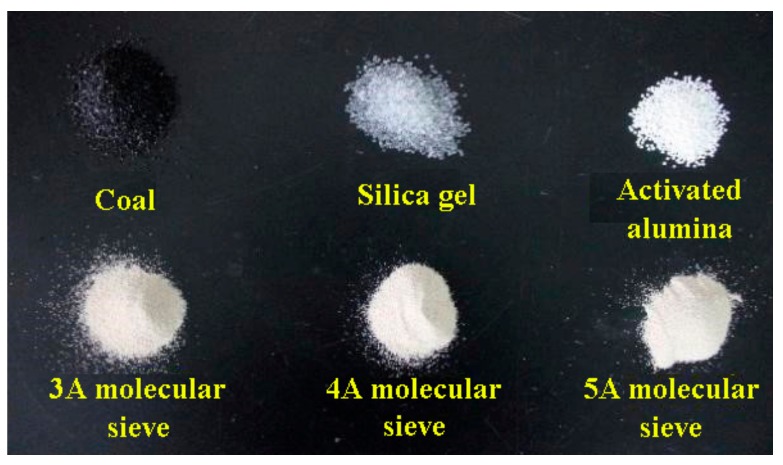
Testing materials selected in the experiment: coal, silica gel, activated alumina, the 3Å molecular sieve, 4Å molecular sieve and 5Å molecular sieve.

**Figure 2 materials-13-00751-f002:**
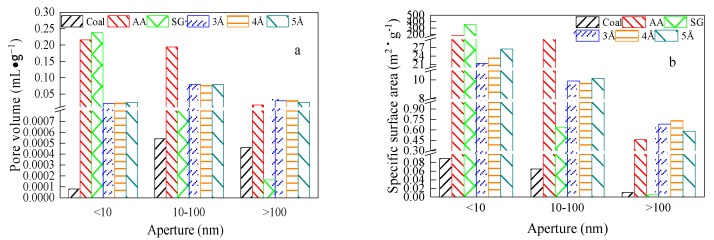
Distribution of pore volume and specific surface area of test materials: (**a**) Pore volume distribution of test materials; (**b**) Specific surface area distribution of test materials.

**Figure 3 materials-13-00751-f003:**
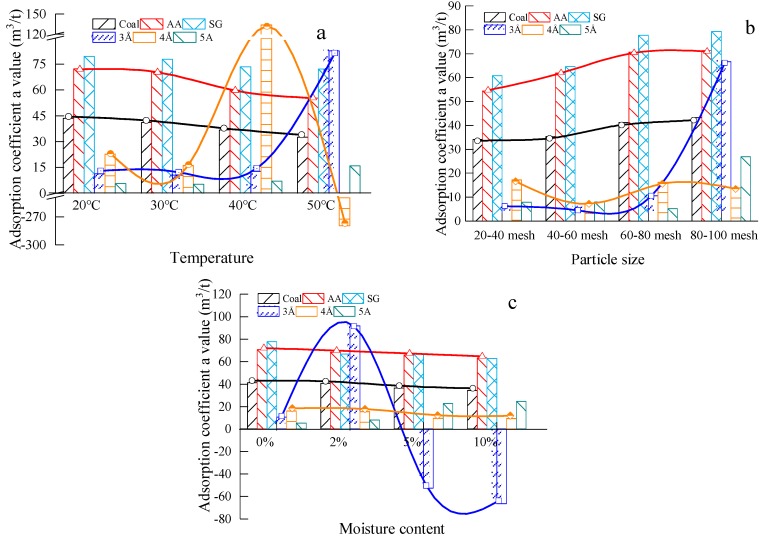
Adsorption coefficient *a* values of testing materials in different factors: (**a**) Adsorption coefficient *a* values of the test materials at different temperatures; (**b**) Adsorption coefficient *a* values of test materials with different particle sizes; (**c**) Adsorption coefficient *a* values of test materials under different moisture contents.

**Figure 4 materials-13-00751-f004:**
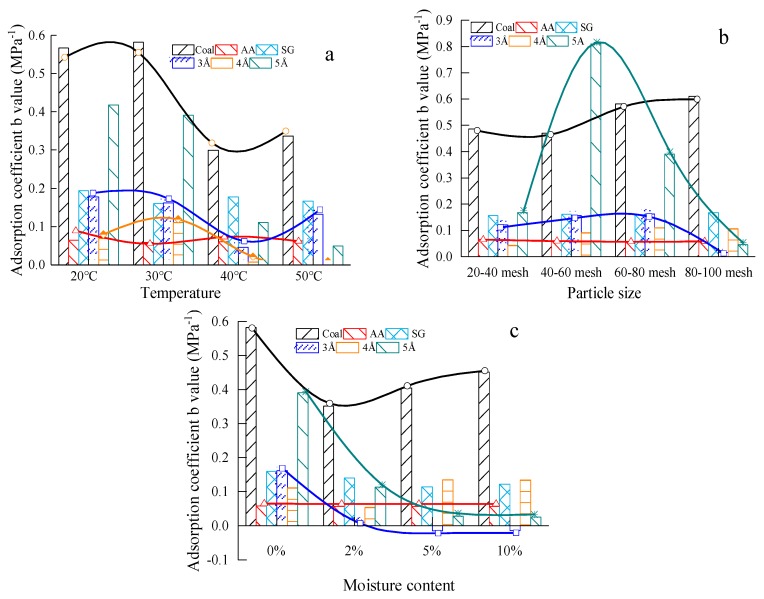
Adsorption coefficient *b* values of testing materials in different factors: (**a**) Adsorption coefficient *b* value of test materials at different temperature; (**b**) Adsorption coefficient *b* value of test materials with different particle size; (**c**) Adsorption coefficient *b* value of test materials under different moisture content.

**Figure 5 materials-13-00751-f005:**
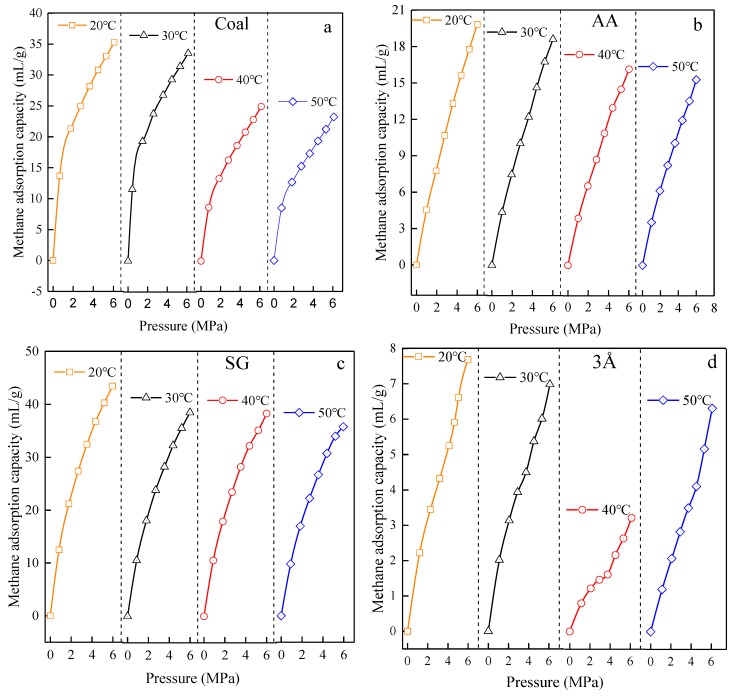
Adsorption isotherms of testing materials in different temperature: (**a**) Aiweiergou 1890# working face coal sample; (**b**) Activated alumina; (**c**) Silica gel; (**d**) 3Å molecular sieve; (**e**) 4Å molecular sieve; (**f**) 5Å molecular sieve.

**Figure 6 materials-13-00751-f006:**
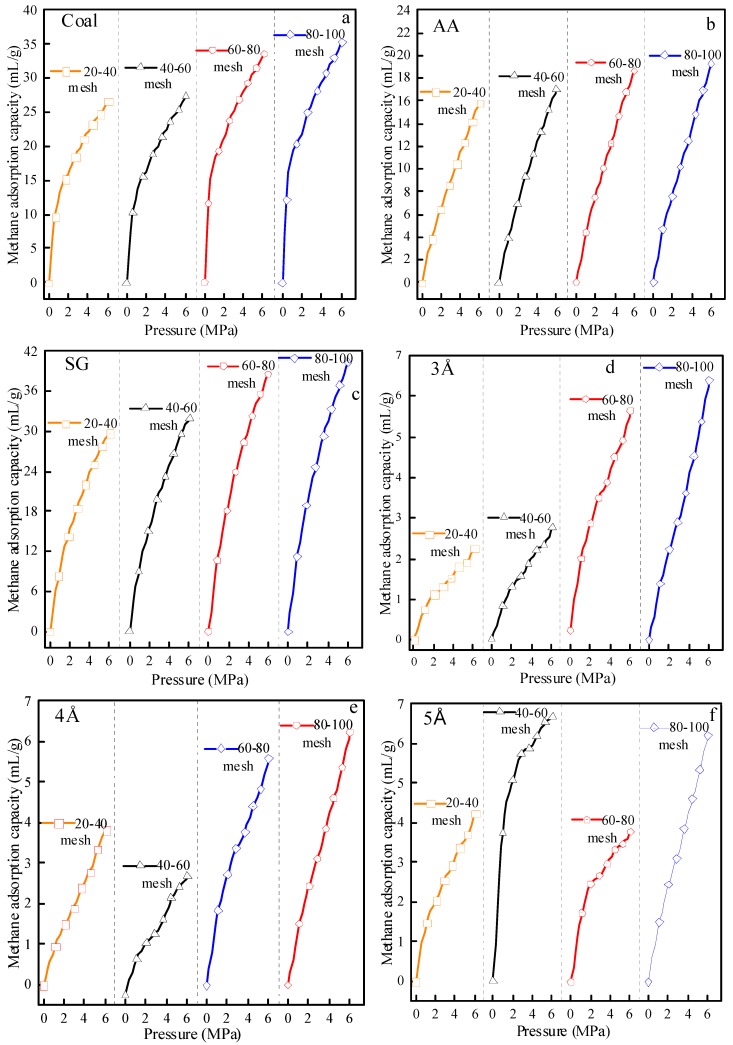
Methane adsorption capacity of testing materials in different particle size: (**a**) Aiweiergou 1890# working face coal sample; (**b**) Activated alumina; (**c**) Silica gel; (**d**) 3Å molecular sieve; (**e**) 4Å molecular sieve; (**f**) 5Å molecular sieve.

**Figure 7 materials-13-00751-f007:**
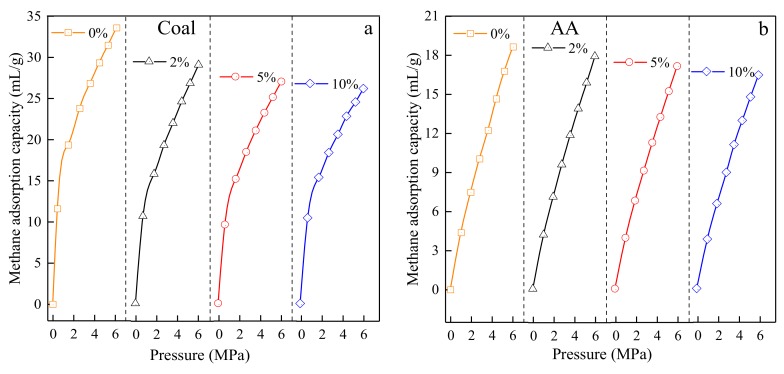
Adsorption isotherms of testing materials in different moisture: (**a**) Aiweiergou 1890# working face coal sample; (**b**) Activated alumina; (**c**) Silica gel; (**d**) 3Å molecular sieve; (**e**) 4Å molecular sieve; (**f**) 5Å molecular sieve.

**Figure 8 materials-13-00751-f008:**
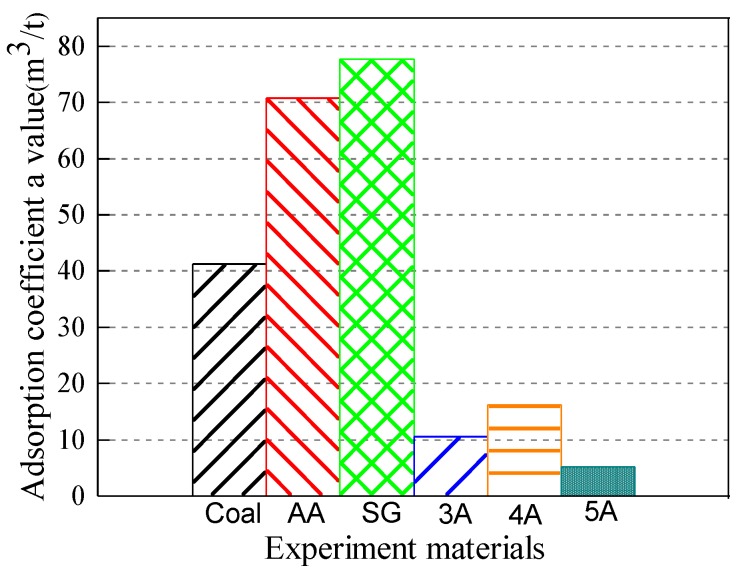
Adsorption coefficient *a* value of testing materials in constant factors.

**Figure 9 materials-13-00751-f009:**
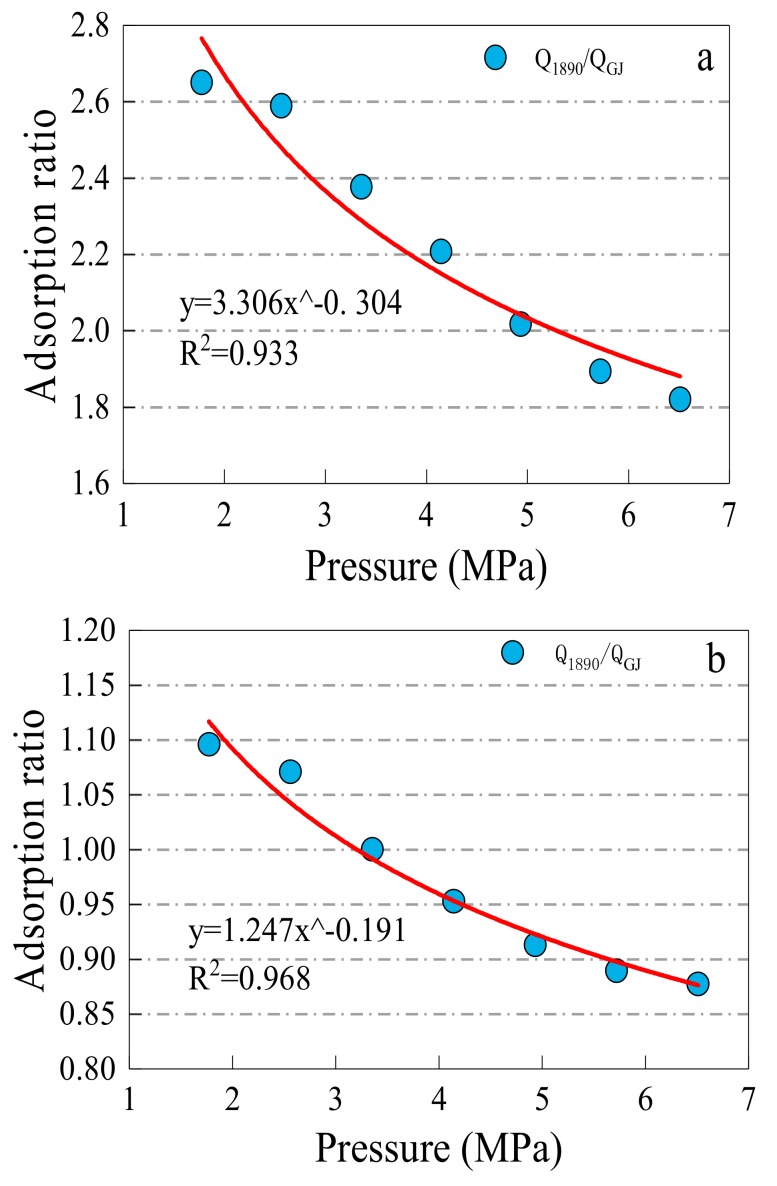
Methane adsorption ratio between adsorption materials and #1890 coal samples in pressure point: (**a**) Q_1890_/ Q_AA_; (**b**) Q_1890_/ Q_SG_. Note: Q1890, QAA, and QSG indicate the methane adsorption capacity of the #1890 working face coal sample of Aiweiergou, the activated alumina and silica gel, respectively.

**Figure 10 materials-13-00751-f010:**
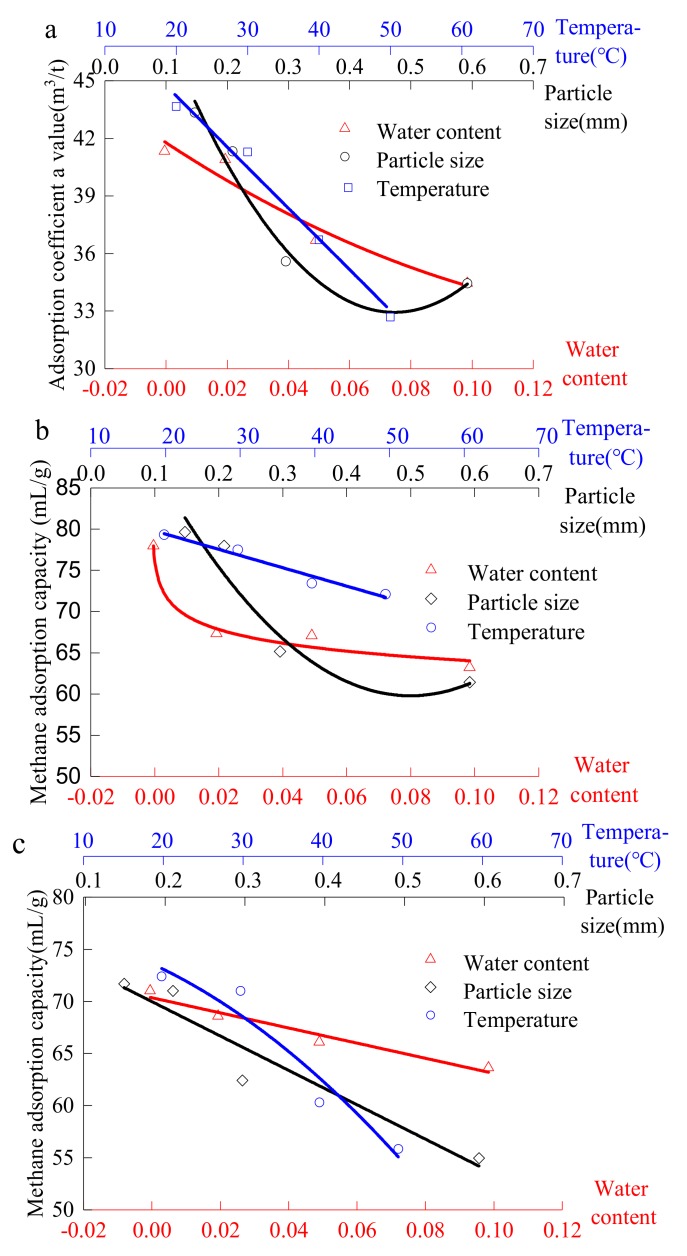
Relationship between the amount of methane adsorption and different factors of testing materials: (**a**) Aiweiergou #1890 working face coal sample; (**b**) Activated alumina; (**c**) Silica gel.

**Table 1 materials-13-00751-t001:** Fitting results of materials under three factors.

Experiment Material		1890	Activated Alumina	Silica Gel	3Å Molecular Sieve	4Å Molecular Sieve	5Å Molecular Sieve
Temperature	20 °C	0.97	0.98	0.998	0.89	0.85	0.28
	30 °C	0.96	0.99	0.99	0.97	0.94	0.99
	40 °C	0.98	0.95	0.99	0.52	0.22	0.71
	50 °C	0.96	0.99	0.99	0.16	0.19	0.74
Particle size	#20–#40 sieve	0.98	0.98	0.99	0.95	0.83	0.97
	#40–#60 sieve	0.97	0.99	0.99	0.96	0.86	1.00
	#60–#80 sieve	0.96	0.98	0.99	0.97	0.94	0.99
	#80–#100 sieve	0.96	0.95	0.99	0.45	0.88	0.88
Moisture content	0%	0.96	0.99	0.99	0.98	0.94	0.97
	2%	0.97	0.98	0.99	0.37	0.27	0.90
	5%	0.97	0.99	0.99	0.52	0.98	0.62
	10%	0.97	0.96	0.99	0.65	0.98	0.71

**Table 2 materials-13-00751-t002:** Fitting relationship between adsorption and every factor.

Experiment Material	Factor	Formula	Fit
Aiweiergou 1890# working face coal sample	Moisture content	Q = 41.697 − 105.665w − 292.827w^2^	0.947
	Particle size	Q = 56.332 − 98.7d + 102.88d^2^	0.964
	Temperature	Q = 51.704 − 0.375t	0.985
Silica gel	Moisture content	Q = 58.185(w + 4.920 × 10^−4^)^−0.038^	0.975
	Particle size	Q = 103.559 − 175.386d + 173.252d^2^	0.925
	Temperature	Q = 84.905 − 0.262t	0.961
Activated alumina	Moisture content	Q = 70.778 − 73.024w	0.961
	Particle size	Q = 76.842 − 38.720d	0.920
	Temperature	Q = 77.272 − 0.061t − 0.008t^2^	0.938

Note: t is temperature, °C; d is diameter, mm; w is moisture content.

**Table 3 materials-13-00751-t003:** Extremum difference analysis of methane adsorption principle of #1890 coal sample.

Numbering	Moisture Content	Particle Size	Temperature
1	41.194	34.196	43.649
2	40.75	35.36	41.194
3	36.475	41.194	36.701
4	34.185	43.254	32.682
Range	7.009	9.058	10.967

**Table 4 materials-13-00751-t004:** Extremum difference analysis of methane adsorption principle of activated alumina.

Numbering	Moisture Content	Particle Size	Temperature
1	70.721	54.405	72.111
2	68.261	61.979	70.721
3	65.709	70.721	59.845
4	63.205	71.399	55.298
Range	7.516	16.994	16.813

**Table 5 materials-13-00751-t005:** Extremum difference analysis of methane adsorption principle of silica gel.

Numbering	Moisture Content	Particle Size	Temperature
1	77.699	60.841	79.524
2	66.826	64.646	77.699
3	66.593	77.699	73.523
4	62.672	79.361	72.179

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
