# Peer review of "Evaluating Methane Adsorption Characteristics of Coal-Like Materials"

_materials, 2020, doi:10.3390/ma13030751_

Round 1
Reviewer 1 Report
The manuscript submitted presents an extensive experimental study to investigate the gas adsorption characteristics of coal seam materials in solid-gas coupling physical simulation experiments. Five different adsorption materials were used, namely: activated alumina, silica gel and molecular sieves with three different sizes. The experiments were conducted under different temperatures, particle sizes and moisture contents. The authors characterize the used materials, the experimental devices and the used methodology. The obtained results are presented and discussed.
The topic that is developed in the study is interesting and important since it deals with solid-gas coupling phenomenon, which is related with the gas drainage at mining sites. The reported results can be very useful to help for the selection of gas-adsorbing materials which can be used for mining safety.
I made some suggestions in order to improve the manuscript. The authors should take the suggestions into account and revise their manuscript.
Comment 1
The manuscript needs to be revised to improve the reading and understanding, as well as to correct typos. Some parts of the article are not easy to understand and some sentences does not make sense. It is recommended that the authors seek for professional help to improve the manuscript on this point.
Comment 2
Avoid to repeat many times the same statement or some key ideas throughout the manuscript. The Manuscript is somewhat extensive and the revision can help to reduce its length.
Comment 3
In Section 2.2 please provide references for the theoretical models.
Comment 4
Almost all figures need to be enlarged for better reading.
Comment 5
Some formatting issues need to be corrected. For instance, the adsorption coefficient “a” is sometimes written in italic, and other times not. Same for other symbols.
Reviewer 2 Report
Title should be modified by mentioning “methane adsorption”. Furthermore, “the methane adsorption” should be also added at the objectives of this investigation (by the end of introduction part). The out - gassing conditions (T and P) must be mentioned. Please give somewhere the preciseness of the pore size determination.
A point needs a discussion on the fact that coal, silica gel and activated alumina present mainly external surfaces made of equilibrium or cleavage faces, contrarily to molecular sieves that present micro-porosities with bottle-like pores. In the first case, the surface energies are distributed but in the second one, the interaction energies are conditioned by the size of access aperture, then by a structural property. In the last case, the desorption energies are the same for most of the molecules. Since the authors refer to Langmuir formalism, it must be considered that this theory is based upon the fact that the surface is energetically homogeneous.
“…4A molecular sieve,…”
“4A” should be “4 Å”.
“…the following five components including active alumina (Al2O3),…”
Please mention the alumina polytype, since there are large structural differences between the species.
“…the adsorption and desorption curves of liquid nitrogen under different pressures were obtained, and…”
The reviewer feels that the adsorption isotherms are not obtained with liquid nitrogen, but with gaseous nitrogen to the boiling temperature of liquid nitrogen. Please check and modify, if necessary.
See also further in the manuscript: “…was measured by liquid nitrogen adsorption…”.
“…(multilayer adsorption theory created by Brunauer, Emmett and Teller) theoretical model and the BJH (aperture distribution model created by Barrett, Joyner and Halenda) theoretical model.”
Please add reference(s) for these two models.
“According to the requirements of the instrument, and suggestions of previous research work [26], five sets of sieves (#20, #40, #60, #80, and #100) were used to prepare aggregates for gas isothermal adsorption test.”
Please check Reference [26]. What is the correlation between the work done in this Reference and the present investigation?
“B.B. Hoodot divided the pores in coal into micropores (< 10 nm),…”
Please add a Reference for this affirmation.
“It is known from Fig. 4(a) that the pore volume of silica gel…”
Fig. 4(a) should be Fig. 3(a). Please check and modify, if necessary.
“Fig. 4(b) shows that the SSA…”
Fig. 4(b) should be Fig. 3(b). Please check and modify, if necessary.
“…researchers use the Langmuir adsorption model to calculate the amount of coal gas adsorption [31-33].”
A rough examination of Reference [31] (see below) showed that this Reference is not devoted to “use the Langmuir adsorption model. Please check and modify.
Therefore, the authors are invited to check all references in text and in the references list for correctness.
“[31] G.M. Kepler, C. Hopfner, J.S. Scroggs, K.J. Bachmann, Simulation of a vertical reactor for high pressure organometallic chemical vapor deposition, Mat Sci Eng B-Solid 57(1) (1998) 9-17.”
“In the coal-soil gas-solid-gas coupling physical similar simulation experiment, the gas…”
This sentence has to be improved (coal-soil gas-solid-gas??)
“And the vanadium force exists in the surface…”
“vanadium force” should probably be: “van der Waals force”. Please check and modify properly.
Figures: the characters are too small for an easy reading. Adsorption data would be given as a function of the logarithm of the relative pressure, since this unit is homogeneous to a chemical potential.
Please add "doi" of the article (in the Reference list), where possible.
Reviewer 3 Report
Review of paper “Evaluating gas adsorption characteristics of coal-like materials” prepared by Pengxiang Zhao, Hui Liu, Chun-Hsing Ho, Shugang Li, Yanqun Liu, Haifei Lin and Min Yan.
The manuscript Materials-709789 reports results of gas adsorption characteristics of coal-like materials including silica, activated alumina, and molecular sieves, as well as the effect of particle size of coal sample, mining temperature, and moisture of coal materials on the adsorption process. The paper reviewed contains a lot of experimental values, but their interpretation should be improved. In my opinion this article should be corrected before it will be considered as worth to publish in Materials. I have some suggestions/questions which authors may consider prior to publication of this work:
Figure 2 should be removed from the manuscript. Photography of ASAP 2020 surface area and porosity analyzer is not needed. The name of the equipment is enough in experimental part. Everyone can check what the analyzer looks like, and the photo itself does not bring anything to work. In section 2.3.1. 3) authors should indicate the kind of the moisture contents percentage (weight, volume?). The section 2.3.2 1) contains the same content as section 2.2. Authors should re-write the text to avoid repetition. The true density and apparent density of coal samples were measured according to the GB/T 115 217-2008 standard (measurement method of true density of coal). This is Chinese standard and should be described in more detail for foreign readers. Did authors check the proposed drying condition? Were vacuum oven at 60 °C for 6 hours sufficient? The adsorption properties depend particle size of coal sample, mining temperature, and moisture of coal materials. Thus in Figure 4 a-c all parameters should be described, for example in Fig 4a – the information for which particle size and moisture content the temperature dependence is presented. The same comments for Figure 5. Explain in more detail the reason for the negative adsorption results (see Fig 4 c): why only once for molecular sieves 3 A and once for 4 A. What influence will the obtained values have on the desired goal of work. In section 3.3.1 the sentence “The gas adsorption capacity of each molecular sieve at different temperatures is different, and the change is not obvious.”, as well as sentence in section 3.3.2 “The gas adsorption capacity of the three kinds of molecular sieve was different under different particle sizes, and the change was not obvious.” are not very scientific and would require more detailed interpretation. In Figure 10 the equations didn’t correspond with the lines. The equations presented in Figure indicate a linear relationship, whereas the lines are not straight. Moreover the information about temperature used in this experiment should be added.Author Response
Please see the attachment.

Reviewer 4 Report
The work of Zhao et al is a systematic study in which different materials are investigated according to their gas adsorption behaviour. The tests are well performed, results are clear and the subsequent discussion is consistent with the results. The work is of importance in the field of gas adsorption materials and accordingly I recommend the work to be published in the materials journals.
Previous to the acceptance, however, I recommend to follow the suggestions I exposed below:
In the conclusions I would introduce a sentence explaining what it has been done in the work to put into a context the reader (it is usual to read the abstract and go directly to the conclusions) In the beginning of section 3.2 I would introduce the explanation provided in the beginning of the subsection 4.1.2, in which the meaning of the coefficients a and b is provided.Additional aspects
Line 21, don’t fully fit a Langmuir…Author Response
Please see the attachment.

Round 2
Reviewer 2 Report
The question of “alumina polytype” was to clarify the alumina phase (gamma-Al2O3 and/or another phase?).
Please be consistent with references writing.
Example (see below): You have to add the author’s name for this reference.
[30] The Changes of pore structure and specific surface area in water treatment residual before and after phosphorus adsorption, Ion Exchange and Adsorption 35(1) (2019) 60-70. DOI: 10.16026/j.cnki.iea.2019010060.
Author Response
Point 1: The question of “alumina polytype” was to clarify the alumina phase (gamma-Al2O3 and/or another phase?).
Response 1: Thank you for your valuable suggestion. The alumina phase is α-Al2O3. I have revised in the manuscript (page 2, line 76).
Point 2: Please be consistent with references writing.
Example (see below): You have to add the author’s name for this reference.
[30] The Changes of pore structure and specific surface area in water treatment residual before and after phosphorus adsorption, Ion Exchange and Adsorption 35(1) (2019) 60-70. DOI: 10.16026/j.cnki.iea.2019010060.
Response 2: Thank you for your suggestion. I have revised in the manuscript.
[30] F. G. Qiu, B. B. Li, K. M. Fu, J. T. Xu, J. L. Wang, The Changes of pore structure and specific surface area in water treatment residual before and after phosphorus adsorption, Ion Exchange and Adsorption 35(1) (2019) 60-70. DOI: 10.16026/j.cnki.iea.2019010060.
Reviewer 3 Report
Review of paper “Evaluating methane adsorption characteristics of coal-like materials” prepared by Pengxiang Zhao, Hui Liu, Chun-Hsing Ho, Shugang Li, Yanqun Liu, Haifei Lin and Min Yan.
The manuscript Materials-709789 after revision still needs correction. In the previous review I noticed not precise captions in Figures 4 and 5 (currently 3 and 4). Authors added to the text only descriptions, i.e. “The adsorption coefficient a of each material prepared at different temperatures (20, 30, 40, and 50°C) and four moisture contents (0%, 2%, 5%, and 10%) were obtained according to the isothermal adsorption test. The test results are shown in Fig. 3.” but this does not explain the results in the figures. For example, in Figure 3 c the adsorption coefficient a value of test materials under different moisture content is presented but there is no information in which temperature the material was prepared or in Fig 3 a for which moisture contents the dependence of temperature was obtained.
Author Response
Point 1: The manuscript Materials-709789 after revision still needs correction. In the previous review I noticed not precise captions in Figures 4 and 5 (currently 3 and 4). Authors added to the text only descriptions, i.e. “The adsorption coefficient a of each material prepared at different temperatures (20, 30, 40, and 50°C) and four moisture contents (0%, 2%, 5%, and 10%) were obtained according to the isothermal adsorption test. The test results are shown in Fig. 3.” but this does not explain the results in the figures. For example, in Figure 3 c the adsorption coefficient a value of test materials under different moisture content is presented but there is no information in which temperature the material was prepared or in Fig 3 a for which moisture contents the dependence of temperature was obtained.
Response 1: Thank you for your valuable suggestion. I have revised in the manuscript (section 3.2.1 and 3.2.2).